# Distinctive Features of Orbital Adipose Tissue (OAT) in Graves’ Orbitopathy

**DOI:** 10.3390/ijms21239145

**Published:** 2020-11-30

**Authors:** Lei Zhang, Anna Evans, Chris von Ruhland, Mohd Shazli Draman, Sarah Edkins, Amy E. Vincent, Rolando Berlinguer-Palmini, D. Aled Rees, Anjana S Haridas, Dan Morris, Andrew R. Tee, Marian Ludgate, Doug M. Turnbull, Fredrik Karpe, Colin M. Dayan

**Affiliations:** 1School of Medicine, Cardiff University, Heath Park, Cardiff CF14 4XN, UK; evansa85@outlook.com (A.E.); VonRuhlandCJ@cardiff.ac.uk (C.v.R.); shazlidraman@gmail.com (M.S.D.); EdkinsS@cardiff.ac.uk (S.E.); ReesDA@cardiff.ac.uk (D.A.R.); TeeA@cardiff.ac.uk (A.R.T.); Ludgate@cardiff.ac.uk (M.L.); DayanCM@cardiff.ac.uk (C.M.D.); 2Wellcome Centre for Mitochondrial Research, Translational and Clinical Research Institute, Faculty of Medical Sciences, Newcastle University, Newcastle NE2 4HH, UK; Amy.Vincent@newcastle.ac.uk (A.E.V.); doug.turnbull@newcastle.ac.uk (D.M.T.); 3The Bio-Imaging Unit, Faculty of Medical Sciences, Newcastle University, Newcastle NE2 4HH, UK; rolando.berlinguer-palmini@newcastle.ac.uk; 4Department of Ophthalmology, Cardiff & Vale University Health Board, Cardiff CF14 4XW, UK; anjana@doctors.org.uk (A.S.H.); dsm@doctors.org.uk (D.M.); 5Oxford Centre for Diabetes, Endocrinology and Metabolism, Radcliffe Department of Medicine, University of Oxford, Oxford OX3 7LE, UK; fredrik.karpe@ocdem.ox.ac.uk; 6NIHR Oxford Biomedical Research Center, OUH Foundation Trust, Oxford OX4 2PG, UK

**Keywords:** Graves’ orbitopathy, orbital adipose tissue, WAT, BAT and BRITE, fatty acid uptake, hyperplasic adipocyte expansion

## Abstract

Depot specific expansion of orbital-adipose-tissue (OAT) in Graves’ Orbitopathy (GO) is associated with lipid metabolism signaling defects. We hypothesize that the unique adipocyte biology of OAT facilitates its expansion in GO. A comprehensive comparison of OAT and white-adipose-tissue (WAT) was performed by light/electron-microscopy, lipidomic and transcriptional analysis using ex vivo WAT, healthy OAT (OAT-H) and OAT from GO (OAT-GO). OAT-H/OAT-GO have a single lipid-vacuole and low mitochondrial number. Lower lipolytic activity and smaller adipocytes of OAT-H/OAT-GO, accompanied by similar essential linoleic fatty acid (FA) and (low) FA synthesis to WAT, revealed a hyperplastic OAT expansion through external FA-uptake via abundant *SLC27A6* (FA-transporter) expression. Mitochondrial dysfunction of OAT in GO was apparent, as evidenced by the increased mRNA expression of uncoupling protein 1 (*UCP1*) and mitofusin-2 (*MFN2*) in OAT-GO compared to OAT-H. Transcriptional profiles of OAT-H revealed high expression of Iroquois homeobox-family (*IRX-3&5*), and low expression in *HOX*-family/*TBX5* (essential for WAT/BAT (brown-adipose-tissue)/BRITE (BRown-in-whITE) development). We demonstrated unique features of OAT not presented in either WAT or BAT/BRITE. This study reveals that the pathologically enhanced FA-uptake driven hyperplastic expansion of OAT in GO is associated with a depot specific mechanism (the *SLC27A6* FA-transporter) and mitochondrial dysfunction. We uncovered that OAT functions as a distinctive fat depot, providing novel insights into adipocyte biology and the pathological development of OAT expansion in GO.

## 1. Introduction

Orbital adipose tissue (OAT) is dysregulated in the disfiguring and sight-threatening disease of the orbit, Graves’ Orbitopathy (GO, also called thyroid eye disease) [1,2]. In the healthy state, OAT acts to cushion and protect the orbital contents, extra-ocular muscles, blood vessels and nerves, in the bony orbit [3]. In GO, the uncontrolled expansion of OAT results in periorbital swelling, pain, redness, proptosis, double vision, and in some cases visual loss, which develops mainly in the context of the associated autoimmune condition, Graves’ disease [1,2]. Previous studies have examined the autoimmune/inflammatory changes in OAT in GO with a focus on activation pathways via two main targeted receptors, the thyrotropin receptor (TSHR) and insulin like growth factor 1 receptor (IGF1R) [4,5,6,7,8]. However, little attention has been directed to basic orbital fat biology and how it changes in patients with GO. 

Uncovering the biological differences between OAT and white adipose tissue (WAT) will enhance our understanding of the pathophysiology of the disease. Known differences include: (i) OAT is a neural crest derived fat depot within the orbit where WAT and brown adipose tissue (BAT) have a mesodermal origin [9,10]; (ii) in contrast to the increasing WAT mass in obesity worldwide, there are no reports of OAT expansion or proptosis in obesity [11]; and (iii) in patients with GO, a marked increased volume of OAT is due to adipogenesis of preadipocytes/fibroblasts (PFs) [12], while WAT in the same individual typically shrinks due to hyperthyroidism [13]. We previously reported depot specific signaling networks involving the PI3K/Akt/mTORC1/FOXOs pathways present in PFs derived from human OAT distinct from WAT [14,15], and these appear to play essential roles in OAT expansion in GO [2]. 

Adipose tissue plays a fundamental role in regulating whole-body energy homeostasis including adipogenesis, lipid uptake, de novo lipogenesis, unsaturated fatty acids (uFAs) conversion and lipolysis, as well as producing regulatory hormones such as adiponectin (*ADIPOQ*) and leptin (*LEP*) [16]. WAT as a ‘flexible energy reservoir’ that supplies free fatty acids (FAs) when needed, while BAT functions to burn FAs to generate cold-induced adaptive thermogenesis [17]. There is a single lipid vacuole in the adipocytes of WAT contrasting to the multi-loculated appearance of BAT [18]. BRITE (BRown in whITE) adipocytes are found in human WAT and BAT depots [19,20]. BAT/BRITE is distinguished from WAT by high expression levels of uncoupling protein 1 (*UCP1*), which dissipates energy as heat from the abundant mitochondria present in adipocytes [17,20]. Adipogenesis is a lineage specific differentiation process that occurs in PFs to form adipocytes. In this process, lipid accumulates intracellularly as triglyceride from the uptake of external FAs and/or de novo lipogenesis from internal carbohydrates. To expand adipose tissue, BAT increases adipocyte number (hyperplasia) [21], while WAT is mainly considered to expand by increasing the size of adipocytes (hypertrophy), although variation in hyperplasia capacity has been observed between different WAT depots [22]. 

OAT shares several features with BAT. For instance, adiponectin overexpression in a murine model showed a selective expansion of fat depots including intrascapular (known to be BAT) and OAT [23]. Furthermore, increased *UCP1* expression in OAT has been observed from a recent TSHR-induced GO mouse model [24]. We have demonstrated that TSHR activation plays an important role in the up-regulation of *UCP1* in both OAT and WAT [19,25], also suggesting that OAT is closely related to BAT. We hypothesize that the unique adipocyte biology of OAT provides the link to its expansion in GO. In this study, we characterize the OAT fat depot, by morphology, lipidomic and transcriptional analysis, to gain further insights into OAT adipocyte biology and possible disease mechanisms of GO.

## 2. Results

### 2.1. Adipocytes Size and Mitochondria Count per Adipocyte

We examined ~100 to 500 adipocytes per ex vivo human adipose sample and observed no gross difference in morphological appearance from WAT and healthy OAT (OAT-H), or OAT from GO patients (OAT-GO) by light microscopy (Figure 1A). In particular, adipocytes from WAT and OAT-H/OAT-GO had a single lipid vacuole, in contrast to the multi-loculated appearance of classical BAT [18]. However, adipocytes from OAT-H/OAT-GO were substantially smaller (by about 50%) compared with WAT (Figure 1B).

Enumeration of mitochondria in each group by electron microscopy (Figure 1C) showed 0 to 25 profiles per adipocyte field of view with no significant difference in the number of mitochondria per adipocyte between WAT and OAT-H (Figure 1D). Interestingly significantly fewer mitochondria per adipocyte were observed in an active OAT-GO sample compared with WAT and OAT-H (Figure 1D). No significant differences of length and width of mitochondria were seen comparing OAT-H, OAT-GO and/or WAT (Figure 1E).

### 2.2. Differences in OAT versus WAT or BAT/BRITE in Marker Genes

Expression of marker genes for WAT and BAT/BRITE was analyzed by QPCR in ex vivo samples from OAT-H, OAT-GO and WAT. Substantially down-regulated *LEP* (Leptin, WAT marker) and little expression of *UCP1* (BAT/BRITE marker) were observed in OAT-H when compared with WAT (Figure 2A,B). Expression of *UCP1* transcripts were increased in OAT-GO compared to OAT-H (4- to 291-fold, *p* = 0.02) reaching levels comparable or greater than WAT (Figure 2B). The mitochondria and BAT/BRITE marker *MFN2* showed a substantially lower expression in OAT-H compared with WAT (Figure 2C), while significantly increased *MFN2* transcripts were observed in OAT-GO compared to OAT-H and reached levels similar to WAT.

### 2.3. FA Composition of Triglyceride Comparing OAT and WAT

Lipidomic analysis by Gas Chromatography (GC) using paired ex vivo OAT-H and WAT from three healthy individuals, revealed a similar level of 18:2 (linoleic), an essential FA which is derived from external sources only (~12% triglyceride composition) (Figure 3A). However, differences in other aspects of FA composition were observed with less unsaturated FAs (uFAs), 16:1 (palmitoleic), 14:1 (myristoleic), 18:ω3 (α-linolenic) and 20:4 (arachidonic), contrasting with higher saturated FA18:0 (stearic) in OAT-H (4.11 ± 0.53%) compared with paired WAT (2.75 ± 0.26%) samples (Figure 3B,C). No differences in other saturated (12:0, 14:0 and 16:0) or unsaturated (18:1, 18:1v, 20:5, 22:5 and 22:6) FAs were seen when comparing WAT with OAT-H (Figure 3).

We also conducted triglyceride FA compositional analysis on non-paired OAT samples from GO patients and observed similar FA composition in OAT-GO compared to OAT-H, including the significantly increased 18:0 and decreased 18:ω3 FAs compared to WAT (Figure 3); however, the differences in expression of 16:1, 14:1 and 20:4 in OAT-GO compared to WAT, were no longer significant.

### 2.4. Whole Transcriptome Analysis of Lipid Metabolisms in OAT Compared to WAT

To better understand OAT, transcriptional profiles were analyzed by RNA sequencing (RNAseq) in ex vivo fat tissue samples of OAT-H (*n* = 4) compared to WAT (*n* = 5). A total of 60,668 genes were in the genome build and 23,766 genes (total counts across all samples > 1) were analyzed (Appendix A). Prior to differential gene expression (DGE) analysis, a good fit normalized count of genes from all samples for DESeq2 model had been shown by shrinkage estimation of dispersion plots (Figure 4A), in which the majority of genes were scattered around the curve of expected dispersion value. At sample-level, Figure 4B showed similar distribution patterns across all samples at lower, median and upper quantiles and counts for outlying features.In DGE analysis comparing OAT-H vs. WAT, we obtained 3790 genes with FDR-*p* significant (<0.05), which has 1746 genes (7.35%) up-regulated and 2044 genes (8.6%) down-regulated. A summarized log2fold change of genes over the mean of normalized counts is shown in the MvA plot (Figure 4C). A PCA-plot at sample-level (Figure 4D) showed a clear separation of samples from OAT-H or WAT that used the top 500 genes (FDR < 0.05) with highest variance (DGE data in Appendix A).DGE analysis comparing OAT-H with WAT confirmed substantially lower expression levels of *MFN2* and *LEP*, as well as a significantly increased expression of *LEP* receptor (*LEPR*) (Appendix A). Lower expression level of *ADIPOQ* (adiponectin) was seen in OAT-H compared with WAT with no difference in *ADIPOQ* receptors (Appendix A). In further analysis, we used a setting of log2fold > 0.5 or <−0.5; FDR-*p* < 0.05, and obtained 3277 genes (1495 up and 1782 down-regulated) for IPA^®^ analysis. The up- or down-regulated gene set of OAT-H versus WAT was also subject to IPA^®^ analysis. Pathway analysis suggested substantially reduced lipid metabolic activity (overlap-*p* < 0.01; z-score < −2) including FA metabolism, lipid uptake, uFAs conversion, lipolysis, lipid synthesis, and oxidation of lipid (e.g., *ACADM*, *ECHS1*) compared with WAT (Appendix A). This evidence for reduced lipid metabolic activity was further supported by IPA^®^ analysis of down-regulated genes. Furthermore, we identified significantly increased expression of the FA transporter, *SLC27A6*, in OAT-H compared with WAT.Targeted QPCR analysis of genes highlighted in the whole transcriptome analysis confirmed substantially lower levels of elements of lipid metabolism including lipid uptake (*LPL*), lipolysis (*LIPE*) and uFAs conversion (*SCD*), and substantially up-regulated *SLC27A6* (FA transporter) in OAT-H, when compared with WAT (Figure 5A–D). *SLC27A6* has been reported as a transporter for long chain FAs uptake (>14 carbons) [27,28].In OAT-GO samples, *LIPE* transcripts (lipolysis) remained low, while variable *LPL* (lipid uptake) and *SCD* (uFAs conversion) expression levels were detected, not significantly different compared to WAT by QPCR analysis (Figure 5A–C). The lack of significant difference of *FASN* (fatty acid synthase) mRNA transcripts between OAT-H, OAT-GO and WAT was confirmed in QPCR analysis (Appendix A), as well as the further increase in *SLC27A6* expression in OAT-GO compared to OAT (Figure 5D).

### 2.5. Signatures of Biological and Signaling Pathways in OAT by IPA^®^ Analysis

Substantially lower expression of the Homeobox (HOX) families (A, B, C, D) were observed in OAT-H compared with WAT (sub-grouped in Appendix A). *HOXA5, HOXA6, HOXC8* and T-box transcription factor 5 (*TBX5*) were among the top most lowly expressed genes in OAT-H and relate to embryonic development of WAT/BAT (highlighted in Appendix A) [9,10]. Upregulation in the Iroquois homeobox gene family (*IRX3* and *IRX5*), which play a role in an early step of neural and adipose tissue development, was observed in OAT-H [29,30]. The top ten up-regulated genes in OAT-H are shown in Appendix A.In IPA^®^, in addition to the changes in lipid metabolisms detailed above, substantially lower activity (overlap-*p* < 0.01; z-score < −2) was seen in pathways of cell movement and molecule transport in OAT-H compared with WAT (Appendix A). In agreement with this, signaling pathways showed substantially lower activity in OAT-H of oxidative phosphorylation (OXPHOS), glycolysis, gluconeogenesis, glycogen degradation and cAMP-mediated signaling (Appendix A). Furthermore, IPA^®^ of down-regulated genes further supported the observed lower metabolic activity and signaling pathways of OAT-H compared with WAT (Appendix A).Certain signaling pathways were substantially activated (overlap-*p* < 0.01; z-score > 2, IPA^®^ analysis) in OAT-H compared with WAT. (A) Activation of Wnt/Ca+ signaling pathway [31] was observed, as indicated by 11 up-regulated genes including Wnt family members (*Wnt5A* and *Wnt5B*) (Appendix A). (B) Components of fibroblast growth factor (FGF) signaling [32] were significantly increased in OAT-H compared to WAT including FGF family and FGF-receptor 2 (*FGFR2*) (Appendix A). Furthermore, we observed significantly increased expression of *IGF2, IGF1R*, IGF-binding proteins (*IGFBP4*&*6*, *IGF2BP2*) but down-regulated *IGF1*, *IGFBP3* and *Akt1* expression [33] (Appendix A). (C) Activation of Sirtuin signaling pathway [34] was observed as indicated by 11 up- and 64 down-regulated genes in IPA^®^ analysis (Appendix A).Additional up-regulated genes in OAT-H compared to WAT included expression of molecules (e.g., neuronal biomarkers, *NRGN* [35] and *SNAP25* [36]) associated with neurogenesis (see Appendix A) including development of neurons, proliferation of neuronal cells, differentiation of neurons, outgrowth of neuritis, neurotransmission and neuritogenesis.Up- and down-regulated expression of growth factors of interest in OAT-H compared with WAT are summarized in Table 1. The up-regulated (range of ~2 to 45-fold, Appendix A) growth factors interfere with signaling pathways including TGFβ/Wnt/Ca+/FGF/Notch/MAPK/PI3K, potentially relevant to a wide range of biological processes of OAT [31,37,38].

## 3. Discussion

### 3.1. OAT Is Distinct from WAT and BAT/BRITE

Our study has highlighted several differences in OAT, which distinguish it from WAT and BAT/BRITE. These are summarized in Table 2. Unlike WAT, OAT has low level *LEP* expression and no hypertrophy but displays the hyperplasia characteristics of BAT/BRITE as evidenced by smaller adipocyte in OAT-H and OAT-GO compared to WAT. However, in contrast to BAT/BRITE, OAT and WAT have a single lipid vacuole, low levels of *UCP1* and *MFN2* expression and similar mitochondria number per adipocyte. Furthermore, OAT is of neural crest origin consistent with this study, identifying substantially lower expression of *HOX*-gene families and *TBX5*, which play an essential role in WAT/BAT development from mesoderm origin [9,10].

The difference between OAT-H and OAT-GO has also been summarized in Table 2, which will be discussed in the following Section 3.2 and Section 3.5. 

### 3.2. Expression of a Distinct FA Transporter in OAT

Our data suggest that there are important differences in the mechanism of adipogenesis in OAT as compared to WAT. Our finding of similar levels of essential linoleic acid and low FA synthesis (de novo lipogenesis) in OAT-H/OAT-GO as in WAT (Table 2) suggest that the depot specific hyperplastic expansion of OAT is FA-uptake driven [22]. This pathologically enhanced FA-uptake in OAT in GO, evidenced by limited lipolysis (*LIPE*) and inducible FA-uptake (*LPL*), may occur via a depot specific FA transport system, for example, *SLC27A6*. SLC27A6 belongs to the FAs transporter family to uptake long-chain FAs functioning in heart or non-tumorigenic breast cells and is absent or only expressed at low levels in WAT [27,28].

### 3.3. OAT Is not Implicated in Maintenance of Energy Balance

Albeit OAT appears to have its stored triglyceride originating externally, it does not store additional triglyceride under conditions of caloric excess in obesity [11]. The low expression levels of leptin and adiponectin, substantially increased leptin receptor and unaltered adiponectin receptor that we observed in the current study indicate that OAT has a role other than maintaining energy balance as do WAT/BAT [17]. In addition to the well-known negative feedback feeding mechanism, leptin via its receptor has direct negative impact on lipid metabolism and is also involved in later neural development, for example, neurogenesis [47,48].

### 3.4. RNAseq Data Provide Additional Insight into Role of IGF-1 Signaling in GO

Previous studies have demonstrated that pathological activation and cross-talk of *TSHR* and *IGF1R* play essential roles in OAT expansion in GO via a depot specific signaling pathway, IGF1R/PI3K/FOXOs [1,2]. This has been emphasized by a recent clinical trial, in which anti-IGF1R therapy (teprotumumab), dramatically reduced proptosis in GO patients [5,7]. The *FGF* signaling pathway has also been shown to be important in OAT expansion in GO [32]. The activation of *FGF/FGFR* increases the expression of *IGF2*, which together play important regulatary roles in MSC via *IGF2/IGF1R* axis [33]. Our current study has demonstrated that *FGFs, FGFR2, IGF2* and *IGF1R* were highly expressed in OAT-H compared with WAT, consistent with the clinical benefits of *IGF1R* inhibition in GO.

### 3.5. Hypothesis of Dysfunction of Mitochondria in OAT Expansion in GO

*TSHR* activation significantly increases *UCP1* expression in OAT, and expansion of OAT and BAT has been reported following overexpression of adiponectin [23,24,25]. Both adiponectin and *UCP1* play important roles in mitochondrial function [49,50]. Our current study showed an activated Sirtuin signaling pathway in OAT-H indicating a potential involvement of mitochondria in OAT metabolism also [34]. Furthermore, we observed pathologically increased expression levels of *UCP1*/*MFN2* in OAT-GO compared to OAT-H. Taken together these data suggest a role for mitochondria in the hyperplasia/adipogenesis of OAT expansion in GO. This hypothesis echoes a recent report of uncontrolled hyperplasia/adipogenesis of BAT expansion linked with dysfunction of mitochondria [51].

### 3.6. Weaknesses and Strengths

The weakness of our study is that many of our results rely on gene transcriptional expression. We performed protein analysis of *UCP1* and mitochondrial markers using immunofluorescence and confocal microscopy; however, low levels of mitochondrial number per adipocyte and expression of relevant markers in WAT/OAT-H/OAT-GO prevented us from obtaining meaningful results.

The strength of our study is the use of human ex vivo adipose tissues, including paired samples of OAT-H and WAT from people free of GO and Graves’ disease, which eliminated individual variation in lipidomic analysis. The main observations from this study have been replicated in more than one independent technique.

### 3.7. Speculations from Current Study

Our study demonstrates a unique adipose tissue with depot specific gene expression of *IRX3* and *IRX5*, which is implicated in the early development of adipose and neural tissue in OAT [29,30]. Apart from maintaining basic lipid metabolism in OAT, transcript profiles were consistent with the potential for neurogenesis, including neuronal biomarkers, for example, high expression of *NRGN* [35] and *SNAP25* [36], *TGFβ* family [42], *JAG1* [43], *MDK* & *PTN* [44], and activation of the Wnt/Ca+ signaling pathway [31]. The innervation of peripheral nerves in WAT/BAT has been established and plays an important role in lipid metabolism and regulation of energy balance [52,53]. Future work to explore further OAT specific neurogenesis and lipid metabolisms would be of great interest.

Studies have shown the multi-differentiation potentials of OAT-PFs as MSC [54]. Our work suggests that OAT-produced growth factors interacting with multiple signaling pathways (TGFβ/Wnt/Ca+/FGF/Notch/MAPK/PI3K) that support repair mechanisms such as adipogenesis, osteogenesis, myogenesis, chondrogenesis, neurogenesis and angiogenesis [31,32,33,37,38,39,40,41,42,43,44,45,46]. Furthermore, we have demonstrated a depot specific triglyceride composition of OAT, for example, stearic acid (18:0 FA), a stable waxy solid fat. This specific triglyceride composition may facilitate its role in supporting constant eye movement [3,55].

The smaller adipocytes, FA-uptake driven adipogenesis of PFs and hyperplastic adipocyte expansion in GO from the current study revealed a key role of the pathological enhanced proliferation of MSC and PFs in OAT in GO. We have previously reported increased proliferation of PFs from OAT-GO compared with OAT-H [56], using the same in vitro models as in this study, our findings agree with that of others [54,57]. Furthermore, our work emphasizes the importance of anti-proliferation therapies, for example, prostaglandin F2α, by targeting MSC/PFs in OAT in GO reviewed in [2]. Future investigation is needed to explore the mechanism triggering MSC/PF proliferation in the autoimmune/inflammatory environment in GO [2] and its possible link with mitochondrial dysfunction [51].

### 3.8. Clinical Impact from Current Study

Our study has demonstrated a depot-specific FA-uptake driven OAT expansion with limited lipolysis in GO. Lipolysis of WAT leads to elevated plasma free FA in Graves’ disease as a consequence of TSHR activation and hyperthyroidism [58,59,60,61,62], which are also major contributors for GO development [1,2]. It is possible that the free FAs from lipolysis of WAT provide resources for the FA-uptake driven adipogenesis of OAT in GO. Interestingly, a 40% decreased GO risk in Graves’ disease was observed with statin therapy but not non-statin cholesterol-lowering medications [63]. Studies have reported that statin lowers plasma cholesterol and also free FAs [64]. Furthermore, up to 3-fold elevated free FAs were observed from smoking individuals [65], and smoking is also a known risk factor of GO [66]. Together with our current study, these data suggest that excessive FAs may contribute to OAT expansion in GO in addition to immune/inflammatory mechanisms [2].

Current therapy for moderate to severe GO comprises high dose intravenous steroids, with some evidence for benefit with immunosuppressive agents and anti- IGF1R therapy [2]. However, for a majority of patients, significant disfigurement persists with a major impact on quality of life and the need for extensive orbital rehabilitative surgery [67]. Our current study suggests that better control of free FAs available for OAT expansion might be beneficial for GO patients in prevention and post-clinical management.

### 3.9. Conclusions

OAT appears to have many features that are different from WAT or BAT/BRITE. We have observed a quiescent fat depot of OAT. When hyperplasia/adipogenesis is triggered as in GO, FA synthesis (de novo lipogenesis) and lipolysis remains low, suggesting there has been abundant uptake of exogenous FAs [1,2]. The observation of high expression of a specific FA transporter, SLC27A6, suggests that the process may be triggered by facilitated FA uptake. Our work has also raised the possibility of mitochondrial dysfunction playing a part in the hyperplastic expansion of OAT in GO as occurs in BAT [51]. Furthermore, we illustrated a unique gene signature of OAT with activated signaling pathways and potential production of necessary growth factors to support depot specific OAT metabolism in its role supporting tissues embedded behind the eye. Further targeted experiments are required to confirm and extend these findings.

## 4. Materials and Methods

All reagents were obtained from Sigma-Aldrich (Dorset, UK) and tissue culture components from Cambrex (Thermo Fisher Scientific, Waltham, MA, USA) unless otherwise stated.

### 4.1. Adipose Tissue Collection and Preparation

Human Adipose Tissue was collected with informed written consent and approved by the South East Wales Research Ethics Committee (30 May 2006) with registry number (06/WSE03/37). WAT (subcutaneous) was from 10 patients undergoing elective open abdominal or breast surgery for non-metabolic conditions. OAT from GO patients (OAT-GO) (*n* = 10) were from 7 inactive GO patients with a CAS (clinical activity score) < 2, 3 active GO with CAS ≥ 3 undergoing decompression surgery. Healthy OAT (OAT-H) from nonGO patients (*n* = 6) were free of thyroid or other inflammatory eye disease who underwent augmented blepharoplasty. Adipose tissues were snap-frozen, kept in liquid nitrogen for later ex vivo analysis (all samples summarized in Table 3).

### 4.2. Adipocytes Analysis in Ex Vivo Adipose Tissues by Light Microscopy

Adipocytes were analyzed from WAT (*n* = 2, total 254 cells), OAT-H (*n* = 2, total 651 cells) and OAT-GO (*n* = 5, 2 active GO (total 591 cells) and 3 inactive GO (total 1480 cells)) (Table 3), adapted from the previously described method [68]. In brief, adipose tissues were thawed in 4% formaldehyde + 0.2% glutaraldehyde in 300mM phosphate buffer (pH 7.4), cut into 1mm thick slices and fixed for 24h. Samples were then post-fixed for 2h in 2% aqueous uranyl acetate followed by full dehydration in isopropanol (50%, 70%, 90% 10 min each, 100% 2 × 15 min) and infiltration with LR White acrylic resin (London Resin Company, Reading, U.K.) (50% in isopropanol 30 min, neat resin 4 × 20 min). Samples placed in resin-filled size 0 gelatine capsules and polymerized overnight at 50 °C. Sections (0.35µm) were cut on an Ultracut E ultramicrotome and stained with 1% toluidine blue. Sections were examined with an Olympus BX51 research light microscope (Olympus Optical Co. Ltd., London, UK) and digital photomicrographs captured with a Zeiss Axiocam and Axiovision software (Carl Zeiss Vision GmbH, Hallbergmoos, Germany). Adipocytes were segmented manually, by drawing around the outer edge of the fat reservoir, and areas calculated using Image J software (v1.51w).

### 4.3. Electronic Microscopy Analysis of Adipocytes and Mitochondria in Ex Vivo Tissues

Thin (100nm) sections of WAT (*n* = 2), OAT-H (*n* = 2) and OAT-GO (*n* = 1, active GO1) (Table 3) were cut on an Ultracut E ultramicrotome onto 200 mesh formvar/carbon-coated grids, stained with lead citrate, and examined with an Hitachi HT7800 TEM at 80kV (Hitachi High Technologies Corporation, Tokyo, Japan) and digital images acquired with a Megaview G3 camera and Radius software (EMSIS GmbH, Muenster, Germany). Mitochondria in each adipocyte were counted and measured (length and width) from WAT (37 adipocytes), OAT-H (39 adipocytes) and OAT-GO (33 adipocytes) using Image J software (v1.51w).

### 4.4. QPCR for Markers of Adipose Tissues of WAT, BAT/BRITE and OAT

The ex vivo tissues have been analyzed using the primers listed in Appendix A. Total RNA was extracted and cDNA were synthesized using standard protocols for QPCR analysis of mRNA expression as described before [69]. QPCR was conducted using Invitrogen^®^ SYBR^®^ Green QPCR SuperMix-UDG on a Stratagene MX 3000 with 50 °C 2 min; 95 °C 2 min; 40 cycles of 15 s 95 °C and 30 s 60 °C. The relative expression ratio of the targeted gene was calculated using the detected Ct value in comparison to a reference gene (housekeeping gene APRT (adenosine phosphoribosyl transferase)) by the standard method [70].

### 4.5. RNA-Seq Sample Preparation and Sequencing

Total RNA was isolated OAT-H (nonGO female individuals, *n* = 4, age 63.8 ± 5.1) and WAT (nonGO female patients, *n* = 5, age 60 ± 5.7) as shown in Table 3, the quality and quantity was assessed using Agilent 2100 Bioanalyzer and a RNA Nano 6000 kit (Agilent Technologies, Santa Clara, CA, USA). A total of 900ng of Total RNA with a RIN value >8 was depleted of ribosomal RNA and the sequencing libraries were prepared using the Illumina^®^ TruSeq^®^ Stranded Total RNA with Ribo-Zero Gold™ kit (Illumina Inc., San Diego, CA, USA). The steps included rRNA depletion and cleanup, RNA fragmentation, 1st strand cDNA synthesis, 2nd strand cDNA synthesis, adenylation of 3′ ends, adapter ligation, PCR amplification (12-cycles) and validation. The manufacturer’s instructions were followed except for the cleanup after the ribozero depletion step where Ampure^®^XP beads (Beckman Coulter, High Wycombe, UK) and 80% Ethanol were used. The libraries were validated using the Agilent 2100 Bioanalyzer and a high-sensitivity kit (Agilent Technologies, Santa Clara, CA, USA) to ascertain the insert size, and the Qubit^®^ (Thermo Fisher Scientific, Waltham, MA, USA) was used to perform the fluorometric quantitation. Following validation, the libraries were normalized to 4nM, pooled together and clustered on the cBot™2 following the manufacturer’s recommendations. The pool was then sequenced using a 75-base paired-end (2 × 75 bp PE) dual index read format on the Illumina^®^ HiSeq2500 in high-output mode according to the manufacturer’s instructions.

### 4.6. RNA-seq Read Alignment and Differential Expression Analysis

Preprocessed RNA-Seq reads were quality checked (QC) using FastQC v0.11.8 (Babraham Institute), then trimmed of the adapter and low quality read ends using Trim Galore! v0.6.0 (Babraham Institute). Trimmed reads were mapped to Gencode GRCh38 release 32 primary assembly [71] using the STAR v2.5.1b 2-pass method [72]. RSeQC (v3.0.1) infer_experiment.py was used to infer library strand specificity from sequence alignment files (.bam). Then read counts were summarized at exon level and aggregated gene level, specifying strandedness, using Subread FeatureCounts (version 2.0.0) [73], where features were defined by the GRCh38 v32 comprehensive primary assembly gene annotation superset. Normalized counts of genes were generated from raw counts by division of the DESeq2 size factor (Table 3) of each sample (DESeq2 v1.24.0) in differential gene expression (DGE) analysis (R/Bioconductor DESeq2 package) [26]. Log2-fold changes of OAT-H compared to WAT were obtained, and the resultant *p*-values were corrected for multiple testing and false discovery using the FDR method of Benjamini–Hochberg. A FDR-corrected *p*-value (FDR-*p*) threshold of less than 0.05 was used as the criteria for the identification of significantly differentially expressed transcripts. Variance stabilizing transformation counts of genes were generated to present a constant variance for visualization analysis of boxplots and PCA plot at sample level using R/Bioconductor DESeq2 package.

### 4.7. Ingenuity^®^ Pathway Analysis (IPA^®^, Qiagen)

DGE data set of OAT-H versus WAT was used for IPA^®^ (version 01-07) with a setting of log2-fold > 0.5 or <−0.5; FDR-*p*-value < 0.05 (Appendix A). The down- or up-regulated gene set was also analyzed separately. IPA Core Analysis was used to investigate the molecule connectivity (networks), biology functions, and involvement in signaling pathways of OAT-H compared to WAT. The biological activities of OAT genes were analyzed by Fisher’s Exact Test, in which an overlap *p*-value (overlap-*p*) < 0.01 is considered as a significant overlap between our data and the known gene functions stored in Ingenuity^®^ Knowledge Base. Z-score was calculated by IPA^®^ reflecting the overall predicted activation state of OAT biological activities (<−2: inhibited, >2: activated). Ratio shows the mapping number of genes from the current study divided by the total number of genes from the same pathway by Ingenuity^®^ Knowledge Base.

### 4.8. Gas Chromatography (GC) Analysis of Triglyceride FAs

Total lipids were extracted from paired OAT-H and WAT from 3 healthy female individuals (age 49 ± 13.7) (Table 3), and also from 3 unpaired OAT from GO female patients (age, 57.3 ± 18), triglyceride separated and analyzed by GC, as previously described [74]. FA concentrations were calculated relative to the internal standard and results were expressed as a mole percentage of triglyceride.

### 4.9. Statistical and Bioinformatics Analysis

Results were analyzed and a *t*-test (normally distributed data) or nonparametric test was used in this study. Differences between groups were analyzed using ANOVA. In all cases *p* < 0.05 was considered significant. Data are presented as mean ± SEM in this study. Specified FDR-*p*-value for RNAseq data analysis and overlap-*p*-value and z score for IPA^®^ analysis have been described above in their sections.

## Figures and Tables

**Figure 1 ijms-21-09145-f001:**
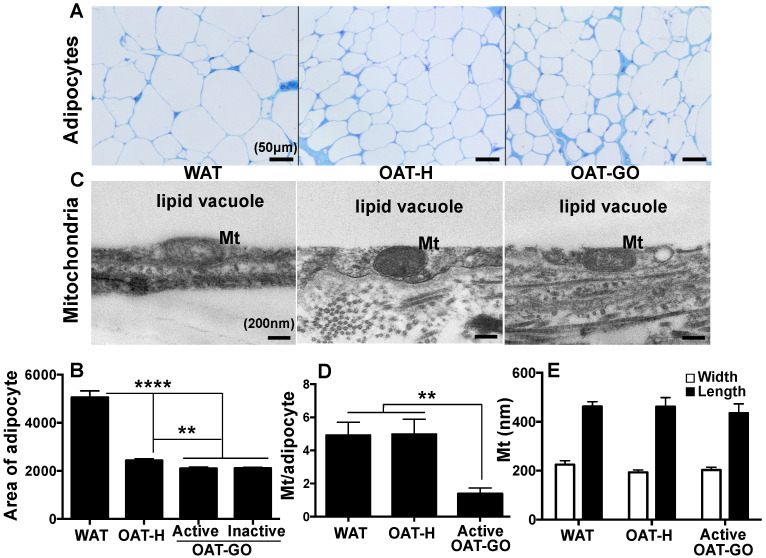
Adipocyte and mitochondria analysis of white-adipose-tissue (WAT), healthy orbital adipose tissue (OAT-H) and OAT- Graves’ Orbitopathy (GO). (**A**) Representative photomicrographs show toluidine blue-stained sections of WAT, OAT from healthy individuals (OAT-H) and OAT from GO patients (OAT-GO) (scale bar 50 µm) by light microscopy; and (**B**) area of adipocytes (µm^2^) was investigated. (**C**) Representative photomicrographs of adipocyte mitochondria (Mt) are shown by electronic microscopy (scale bar 200 nm); (**D**) numbers of mitochondria (Mt) per adipocyte, (**E**) length and width (nm) of mitochondria from each study group were analyzed. Histograms = mean ± SEM of all samples studied, *p* value indicated in the figure, ** as *p* < 0.01, **** as *p* < 0.0001.

**Figure 2 ijms-21-09145-f002:**
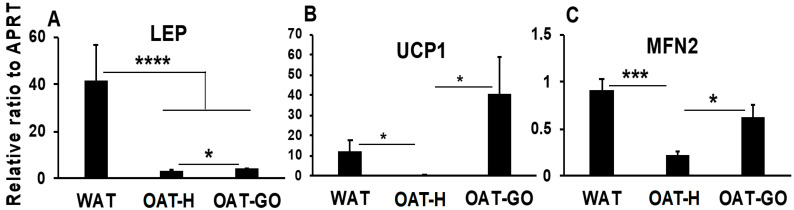
OAT is neither WAT nor brown adipose tissue (BAT)/BRown-in-whITE (BRITE). (**A**–**C**), QPCR analysis of OAT-H (*n* = 4), OAT-GO (*n* = 9) and WAT (*n* = 6) ex vivo fat. *LEP* (WAT marker) and *UCP1* (BAT/BRITE marker) were analyzed using the relative ratio of gene expression by QPCR, and housekeeper gene *APRT* served as the reference gene. *MFN2* (mitochondrial/BAT/BRITE marker) was analyzed by standard PCR, densitometry was measured and corrected to the house-keeping gene (*APRT*), *MFN2* expression relative to *APRT* is presented. Histograms = mean ± SEM of all samples studied. * *p* < 0.05; *** *p* ≤ 0.005, **** *p* ≤ 0.0006.

**Figure 3 ijms-21-09145-f003:**
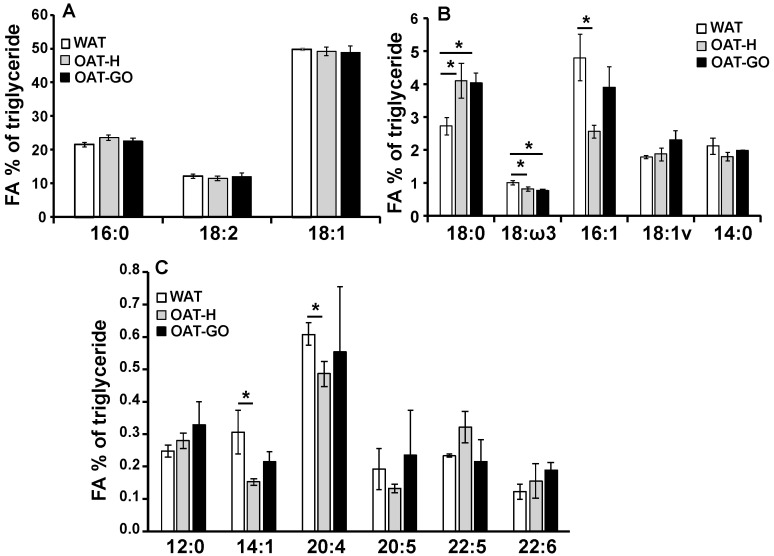
Depot specific fatty acid (FA) composition in OAT. Triglyceride composition of paired ex vivo fat tissues of OAT-H and WAT from three female individuals, also 3 unpaired ex vivo OAT-GO from female GO patients were analyzed by gas chromatography. Data presented as FAs percentage of triglyceride. (**A**) FAs of 16:0, 18:1, 18:2; (**B**) FAs of 14:0, 16:1, 18:0, 18:1v, 18:ω3; (**C**) FAs of 12:0, 14:1, 20:4, 20:5, 22:5, 22:6. Histograms = mean ± SEM of all samples studied. * *p* < 0.05.

**Figure 4 ijms-21-09145-f004:**
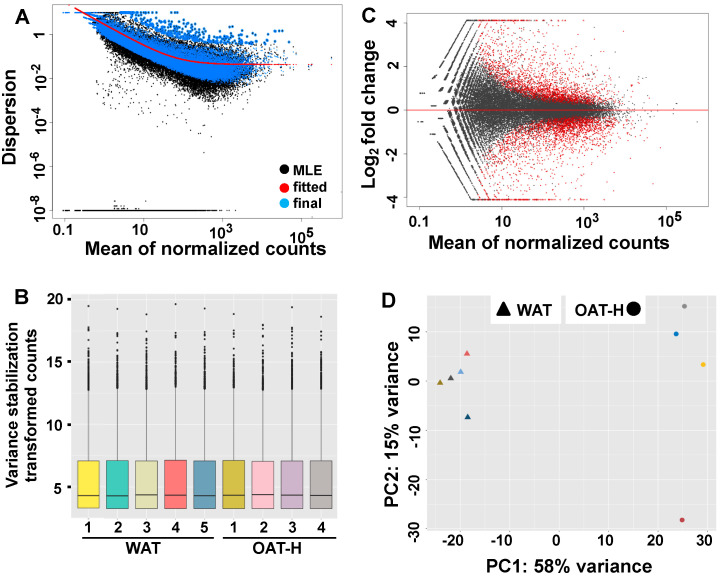
Transcriptional profiles of OAT comparing WAT. Differential gene expression (DGE) analysis was performed using a DESeq2 model generated from ex vivo female samples of OAT-H (*n* = 4) and WAT (*n* = 5) (R/Bioconductor DESeq2 package). (**A**) Shrinkage estimation of dispersion plots provided a visual means of examining dispersion estimates of genomic features relative to average expression strength [26]. Black points represent per gene gene-wise maximum likelihood estimates (MLE). The red curve represents the overall trend of the dispersion to mean dependence. Blue points represent gene-wise estimates following shrinkage towards the mean (correction for noise). Black points circled in blue are dispersion outliers that are not shrunk towards the DESeq2 fitted model. (**B**) Sample level box plots provided a visual means of comparing the distributions of counts between samples, and shown lower, median and upper quantiles and counts for outlying features. (**C**) MvA plot shows the log2fold changes of OAT-H vs. WAT attributable to a given variable of the mean of normalized counts for all samples. Red points indicate a variable with an FDR-*p*-value < 0.1, and points falling outside of the plotting area are depicted as open triangles pointing either up or down. (**D**) A PCA plot was summarized at the sample-level following the exclusion of non-informative features (total counts across all samples < 1) for the top 500 genes in terms of highest variance (Appendix A).

**Figure 5 ijms-21-09145-f005:**
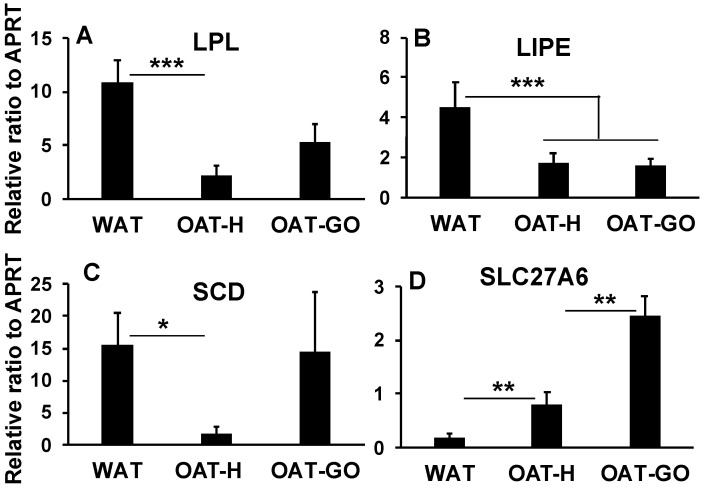
QPCR analysis of lipid metabolisms of OAT compared with WAT. Total RNA was isolated from OAT-H (*n* = 4), OAT-GO (*n* = 9) and WAT (*n* = 6) ex vivo fat tissues. *LPL* (**A**), *LIPE* (**B**), *SCD* (**C**) and *SLC27A6* (**D**) were analyzed using relative ratio of gene expression by QPCR, and housekeeper gene *APRT* served as the reference gene. Histograms = mean ± SEM of all samples studied. * *p* < 0.05; ** *p* ≤ 0.02; *** *p* ≤ 0.005.

**Table 1 ijms-21-09145-t001:** Up- and down-regulated growth factors in OAT-H compared to WAT.

Up-Regulated Growth Factor (Multi-Function 1)	Known Function/Pathway
Angiopoietin (*ANGPT4*) [39]	angiogenesis
Osteoglycin (*OGN*) [40]	bone formation
Neural EGF-like protein (*NELL1*) [41]	bone formation
*TGFβ* family ^1^ [42] (*TGFB2, BMP3, GDF-6,7,11*) and Latent *TGFβ* binding protein ^1^ (*LTBP4*)	*TGFβ* signaling pathway
Jagged 1 protein (*JAG1*) ^1^ [43]	Notch signaling pathway
Midkine (*MDK*) ^1^, Pleiotrophin (*PTN*) ^1^ [44]	multi-signaling pathway
Norrin cystine knot growth factor (*NDP*) ^1^ [45]	Wnt-signaling pathway
Hepatocyte growth factor (*HGF*) ^1^ [46]	HGF/Met signaling pathway
*FGF-(7,9,10,17&18)*^1^ via FGFR [32]	multi-signaling pathway
*IGF2*^1^ via IGF1R/IGF2R [33]	multi-signaling pathway
**Down-regulated growth factors as below**	
*TGFA*, *BMP6*, *GDF10*, *IGF1*, *FGF-2,8&13*, *VEGF-B&D*, *GMFG*, *JAG2*, *LEFTY2*, *LEP* and *PDGF-B&C*	

^1^ Indicated in the table as functioning in multi-cellular processes, e.g., mesenchymal stem cell (MSC) proliferation and differentiation.

**Table 2 ijms-21-09145-t002:** Summary of key features of WAT and BAT/BRITE in OAT from this study.

	OAT-H	OAT-GO	WAT	BAT/BRITE [17,20]
Hypertrophy (Figure 1)	-	-	+++	-
*LEP* (WAT marker) (Figure 2)	---	---	+++	---
Hyperplasia (Figure 1)	+++	+++	-	+++
Lipid vacuole (Figure 1)	-	-	-	+++
Mt number/adipocyte (Figure 1)	-	---	-	+++
*UCP1* (BAT marker) (Figure 2)	---	-	-	+++
*MFN2* (Mt marker) (Figure 2)	---	-	-	+++
*LIPE* (Lipolysis, Figure 5)	-	-	+++	
*LPL* (FA uptake, Figure 5)	-	+++	+++	
*SCD* (uFAs conversion, Figure 5)	---	-	-	
*FASN* (FA synthesis, Appendix A)	-	-	-	
Linoleic acid (essential FA, Figure 3)	+++	+++	+++	
de novo lipogenesis (*SCD*, Linoleic acid and *FASN*)	-	-	-	
Stearic acid (Figure 3)	+++	+++	-	
*SLC27A6* (transporter) (Figure 5)	+++	+++	-	

Key findings of molecular, lipid profile and function differences in OAT have been shown in Table 2, Mt, mitochondria; ‘-’ low level (or single vacuole), ‘---’ lower level, ‘+++’ high level (or multi vacuole)).

**Table 3 ijms-21-09145-t003:** Ex vivo samples from WAT, OAT-H and OAT-GO used for this study.

ID	Sex	Age	GO Status	Analysis
OAT-H1	F	51	no	RNAseq(1.06)
OAT-H2	F	64	no	RNAseq(0.97)
OAT-H3	F	64	no	RNAseq(0.87)/Lipidomic/adipocyte
OAT-H4	F	76	no	RNAseq(1.09)/adipocyte
OAT-H5	F	37	no	Lipidomic
OAT-H6	F	46	no	Lipidomic
OAT-GO1	F	47	active	Lipidomic/QPCR/adipocyte
OAT-GO2	M	49	active	QPCR/adipocyte
OAT-GO3	M	78	active	QPCR
OAT-GO4	F	16	inactive	Lipidomic/QPCR
OAT-GO5	F	56	inactive	Lipidomic/QPCR
OAT-GO6	M	45	inactive	QPCR
OAT-GO7	M	60	inactive	adipocyte
OAT-GO8	F	55	inactive	adipocyte
OAT-GO9	M	29	inactive	adipocyte/PCR
OAT-GO10	F	47	inactive	QPCR
WAT1	F	72	no	RNAseq(1.02)/QPCR
WAT2	F	72	no	RNAseq(0.97)/QPCR
WAT3	F	44	no	RNAseq(1.00)/QPCR
WAT4	F	62	no	RNAseq(1.22)/QPCR
WAT5	F	50	no	RNAseq(0.95)/QPCR
WAT6	F	37	no	Lipidomic/QPCR/adipocyte
WAT7	F	64	no	Lipidomic/QPCR/adipocyte
WAT8	F	46	no	Lipidomic
WAT9	M	49	no	QPCR
WAT10	M	54	no	QPCR

Patients age, sex and GO status are shown in the table. The types of analysis have been shown for each sample as adipocyte, QPCR, lipidomic, and RNAseq analysis is followed by DESeq2 size factor in brackets.

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
