# Peer review of "Distinctive Features of Orbital Adipose Tissue (OAT) in Graves’ Orbitopathy"

_ijms, 2020, doi:10.3390/ijms21239145_

Round 1

Reviewer 1 Report

In this study, Authors characterize the orbital-adipose-tissue (OAT) fat depot to gain further insights into OAT adipocyte biology and possible disease mechanisms of Graves’ Orbitopathy (GO). This topic is very crucial, while understanding of the role of OAT, changes during active phase of GO would be used in practical manner in proposal of new medicine for therapy of GO. I suggest checking again all manuscript for correction of all abbreviations and give all necessary explanations which are missed in some places (e.g. line 22 “linoleic FA”: should be “linoleic fatty acid (FA)”.

Authors compare OAT and white-adipose-tissue (WAT) in light/electron-microscopy, lipidomic and transcriptional analysis. They use OAT from material achieved during decompression (OAT-GO) and from healthy subjects (OAT from non GO patients who underwent blepharoplasty. The authors sometimes use the term OAT interchangeably in context patients with or without thyroid disease. I suggest changing the abbreviation of material from healthy subjects to make it clear. OAT-GO should be used in every place when Authors analyze material from GO, and OAT from patients without thyroid disease should have different abbreviation (OAT-H?; OAT-C: OAT-control?). Probably this is explanation why I don’t know which OAT is described in line 98: “healthy OAT in OAT-GO patients with GO, with inactive GO?

Could Authors describe separately in results and discussion difference between OAT-GO and OAT from control group. This would be interesting.

I suggest improving the part 3.7 Speculations from current study. This part might be especially interesting for clinicians.   

Author Response

We would like to thank the reviewers for their insightful comments, for recognising that the studies are of value for the understanding of depot specific role of orbital adipose tissue in Graves’ Orbitopathy (GO). We hope that the revised manuscript with substantial editorial modifications will now be acceptable for publication.

Rebuttal for ijms-1002217

Ethics information for human adipose tissue collection has been added in Methods Section in Line 365 “Human Adipose Tissue was collected with informed written consent and approved by the South East Wales Research Ethics Committee (30/05/2006) with registry number (06/WSE03/37).”

(Reviewers comments are in italics, while our response follows after each point)

Reviewer 1: Comments and Suggestions for Authors

In this study, Authors characterize the orbital-adipose-tissue (OAT) fat depot to gain further insights into OAT adipocyte biology and possible disease mechanisms of Graves’ Orbitopathy (GO). This topic is very crucial, while understanding of the role of OAT, changes during active phase of GO would be used in practical manner in proposal of new medicine for therapy of GO. I suggest checking again all manuscript for correction of all abbreviations and give all necessary explanations which are missed in some places (e.g. line 22 “linoleic FA”: should be “linoleic fatty acid (FA)”.

We thank the reviewer for recognising the importance of our study: we have checked the manuscript to ensure that all abbreviations are defined, including in figure legends and have provided a list of abbreviations used.

Authors compare OAT and white-adipose-tissue (WAT) in light/electron-microscopy, lipidomic and transcriptional analysis. They use OAT from material achieved during decompression (OAT-GO) and from healthy subjects (OAT from non GO patients who underwent blepharoplasty. The authors sometimes use the term OAT interchangeably in context patients with or without thyroid disease. I suggest changing the abbreviation of material from healthy subjects to make it clear. OAT-GO should be used in every place when Authors analyze material from GO, and OAT from patients without thyroid disease should have different abbreviation (OAT-H?; OAT-C: OAT-control?). Probably this is explanation why I don’t know which OAT is described in line 98: “healthy OAT in OAT-GOà patients with GO, with inactive GO?

We apologise for the confusion and have followed your suggestions throughout the text, tables and figures. We now refer to OAT from healthy individuals as OAT-H e.g. lines 88-92 in the revised manuscript “…in morphological appearance from WAT and heathy OAT (OAT-H), or OAT from GO patients (OAT-GO) by light microscopy (Fig 1A). In particular, adipocytes from WAT and OAT-H/OAT-GO had a single lipid vacuole, in contrast to the multi-loculated appearance of classical BAT [18]. However, adipocytes from OAT-H/OAT-GO were substantially smaller (by about 50%) compared with WAT (Fig 1B).”

We have made clarified of line 98 (line 103 in revision) “Expression of UCP1 transcripts were increased in OAT-GO compared to OAT-H (4- to 291-fold, p=0.02) reaching levels comparable or greater than WAT”; also as line 106 “while significantly increased MFN2 transcripts were observed in OAT-GO compared to OAT-H and reached levels similar to WAT.”

Could Authors describe separately in results and discussion difference between OAT-GO and OAT from control group. This would be interesting.

We have summarized the difference between OAT-GO and OAT-H in Table 2 and discussed in section 3.2. and 3.5.. To be clearer, we added in lines 258-259 “The difference between OAT-H and OAT-GO has also been summarized in Table 2, which will be discussed in the following sections 3.2. and 3.5.”.

I suggest improving the part 3.7 Speculations from current study. This part might be especially interesting for clinicians.

We have improved this section with underline in lines 307-3731 “Our study demonstrates a unique adipose tissue with depot specific gene expression of IRX3 and IRX5, which is implicated in the early development of adipose and neural tissue in OAT [29, 30]. Apart from maintaining basic lipid metabolism in OAT, transcript profiles were consistent with the potential for neurogenesis, including high expression of neuronal biomarkers, e.g. NRGN [35] & SNAP25 [36], TGFb family [42], JAG1 [43], MDK & PTN [44], and activation of the Wnt/Ca+ signaling pathway [31]. The innervation of peripheral nerves in WAT/BAT has been established and plays an important role in lipid metabolism and regulation of energy balance [52, 53]. Future work to explore further OAT specific neurogenesis and lipid metabolisms would be of great interest.

Studies have shown the multi-differentiation potentials of OAT-PFs as MSC [54]. Our work suggests that OAT-produced growth factors interacting with multiple signaling pathways (TGFb/Wnt/Ca+/FGF/Notch/MAPK/PI3K) that support repair mechanisms such as adipogenesis, osteogenesis, myogenesis, chondrogenesis, neurogenesis and angiogenesis [31-33, 37-46]. Furthermore, we have demonstrated a depot specific triglyceride composition of OAT, e.g. stearic acid (18:0 FA), a stable waxy solid fat. This specific triglyceride composition may facilitate its role in supporting constant eye movement [3, 55].

Also added section 3.8. Clinic impact from current study (lines 332-349)

Our study has demonstrated a depot-specific FA-uptake driven OAT expansion with limited lipolysis in GO. Elevated plasma free FA level occurs in Graves’ disease via lipolysis of WAT [58], and particularly due to TSHR activation and hyperthyroidism [59-62], which are also major contributors for GO development [1, 2]. It is possible that the free FAs from lipolysis of WAT provide resources for the FA-uptake driven adipogenesis of OAT in GO. Interestingly, a 40% decreased GO risk in Graves’ disease was observed with statin therapy but not non-statin cholesterol-lowering medications [63]. Studies have reported that statin lowers plasma cholesterol and also free FAs [64]. Furthermore, up to 3-fold elevated free FAs were observed from smoking individuals [65], and smoking is also a known risk factor of GO [66]. Together with our current study, these data suggest that excessive FAs may contribute to OAT expansion in GO in addition to immune/inflammatory mechanisms [2].

Current therapy for moderate to severe GO comprises high dose intravenous steroids, with some evidence for benefit with immunosuppressive agents and anti- IGF1R therapy [2]. However, for a majority of patients significant disfigurement persists with major impact on quality of life and the need for extensive orbital rehabilitative surgery [67]. Our current study suggests that better control of free FAs available for OAT expression might be beneficial for GO patients in prevention and post-clinical management.

Reviewer 2 Report

In the present manuscript the authors have shown data in an effort to explain the role of depot specific expansion of OAT in relation to Graves’ Orbitopathy (GO). The authors have taken advantage of many useful techniques covering the area of molecular biology, qPCR, RNASeq and lipidomics. Though the manuscript looks interesting, in its present form the way it has been written and the data have been arranged does not align very well with the focus of the study. The explanation of the experimental data appears very shallow.

  1. The manuscript appears a platform of presenting too much experimental data obtained from different experimental techniques with no proper explanations why those experiments have been done and what hypothesis the authors have made to test.
  1. All the experimental protocols appear very shallow in describing the methods, explaining the data and their functional implications.
  1. The target genes that have been studied in the present manuscript for the qPCR analysis has not been described properly. I don’t see any information for their accession number, primer pair used and how the qPCR has been performed (cycles and temp) and the experimental data have been analyzed.
  1. Some figures (Figs 2,3 and 5) do not have any Y axes legends.
  1. Ex vivo, in vitro used throughout the manuscript need to be italicized.

Author Response

We would like to thank the reviewers for their insightful comments, for recognising that the studies are of value for the understanding of depot specific role of orbital adipose tissue in Graves’ Orbitopathy (GO). We hope that the revised manuscript with substantial editorial modifications will now be acceptable for publication.

Rebuttal for ijms-1002217

Ethics information for human adipose tissue collection has been added in Methods Section in Line 365 “Human Adipose Tissue was collected with informed written consent and approved by the South East Wales Research Ethics Committee (30/05/2006) with registry number (06/WSE03/37).”

Reviewer 2: Comments and Suggestions for Authors

In the present manuscript the authors have shown data in an effort to explain the role of depot specific expansion of OAT in relation to Graves’ Orbitopathy (GO). The authors have taken advantage of many useful techniques covering the area of molecular biology, qPCR, RNASeq and lipidomics. Though the manuscript looks interesting, in its present form the way it has been written and the data have been arranged does not align very well with the focus of the study. The explanation of the experimental data appears very shallow.

Our study contributes to the study of adipocyte biology and paves a new way to investigate the uncontrolled OAT expansion in GO, which has been clarified as detailed below.  

We have made changes to improve our manuscript as also recommended by reviewer 1, e.g.

We now refer to OAT from healthy individuals as OAT-H throughout the text, tables and figures e.g. lines 89-92 in the revised manuscript “…in morphological appearance from WAT and heathy OAT (OAT-H), or OAT from GO patients (OAT-GO) by light microscopy (Fig 1A). In particular, adipocytes from WAT and OAT-H/OAT-GO had a single lipid vacuole, in contrast to the multi-loculated appearance of classical BAT [18]. However, adipocytes from OAT-H/OAT-GO were substantially smaller (by about 50%) compared with WAT (Fig 1B).”

We have made changes of 3.7 discuss section (lines 307-331) with underline to draw more attention to broad reader “Our study demonstrates a unique adipose tissue with depot specific gene expression of IRX3 and IRX5, which is implicated in the early development of adipose and neural tissue in OAT [29, 30]. Apart from maintaining basic lipid metabolism in OAT, transcript profiles were consistent with the potential for neurogenesis including high expression of neuronal biomarkers, e.g. NRGN [35] & SNAP25 [36], TGFb family [42], JAG1 [43], MDK & PTN [44], and activation of the Wnt/Ca+ signaling pathway [31]. The innervation of peripheral nerves in WAT/BAT has been established and plays an important role in lipid metabolism and regulation of energy balance [52, 53]. Future work to explore OAT further specific neurogenesis and lipid metabolisms would be of great interest.

Studies have shown the multi-differentiation potentials of OAT-PFs as MSC [54]. Our work suggests that OAT-produced growth factors interacting with multiple signaling pathways (TGFb/Wnt/Ca+/FGF/Notch/MAPK/PI3K) that support repair mechanisms such as adipogenesis, osteogenesis, myogenesis, chondrogenesis, neurogenesis and angiogenesis [31-33, 37-46]. Furthermore, we have demonstrated a depot specific triglyceride composition of OAT, e.g. stearic acid (18:0 FA), a stable waxy solid fat. This specific triglyceride composition may facilitate its role in supporting constant eye movement [3, 55].

Also added section 3.8. Clinic impact from current study (lines 332-349)

Our study has demonstrated a depot-specific FA-uptake driven OAT expansion with limited lipolysis in GO. Elevated plasma free FA level occurs in Graves’ disease via lipolysis of WAT [58], and particularly due to TSHR activation and hyperthyroidism [59-62], which are also major contributors for GO development [1, 2]. It is possible that the free FAs from lipolysis of WAT provide resources for the FA-uptake driven adipogenesis of OAT in GO. Interestingly, a 40% decreased GO risk in Graves’ disease was observed with statin therapy but not non-statin cholesterol-lowering medications [63]. Studies have reported that statin lowers plasma cholesterol and also free FAs [64]. Furthermore, up to 3-fold elevated free FAs were observed from smoking individuals [65], and smoking is also a known risk factor of GO [66]. Together with our current study, these data suggest that excessive FAs may contribute to OAT expansion in GO in addition to immune/inflammatory mechanisms [2].

Current therapy for moderate to severe GO comprises high dose intravenous steroids, with some evidence for benefit with immunosuppressive agents and anti- IGF1R therapy [2]. However, for a majority of patients significant disfigurement persists with major impact on quality of life and the need for extensive orbital rehabilitative surgery [67]. Our current study suggests that better control of free FAs available for OAT expression might be beneficial for GO patients in prevention and post-clinical management.

1. The manuscript appears a platform of presenting too much experimental data obtained from different experimental techniques with no proper explanations why those experiments have been done and what hypothesis the authors have made to test.

We have added in abstract line 18 “We hypothesize that the unique adipocyte biology of OAT facilitates its expansion in GO.”; also in introduction lines 80-83 as “We hypothesize that the unique adipocyte biology of OAT provides the link to its expansion in GO. In this study, we characterize the OAT fat depot, by morphology, lipidomic and transcriptional analysis, to gain further insights into OAT adipocyte biology and possible disease mechanisms of GO.”

2. All the experimental protocols appear very shallow in describing the methods, explaining the data and their functional implications.

We apologise for the oversight – probably the result of these protocols being ‘routine’ in our lab. We have expanded the descriptions to provide vital information and also added references, which can direct readers to more detailed protocols.

3. The target genes that have been studied in the present manuscript for the qPCR analysis has not been described properly. I don’t see any information for their accession number, primer pair used and how the qPCR has been performed (cycles and temp) and the experimental data have been analyzed.

Again we apologise for the omissions: all of the information (accession number, primer sequence and exon location) is provided in supplemental table S1. QPCR method is clarified in lines 396-404 with added new reference [70] “The ex vivo tissues have been analyzed using the primers listed in supplemental Table S1. Total RNA was extracted and cDNA were synthesized using standard protocols for QPCR analysis of mRNA expression as described before [69]. QPCR was conducted using Invitrogen® SYBR® Green QPCR SuperMix-UDG on a Stratagene MX 3000 with 50 oC 2 min; 95 oC 2 min; 40 cycles of 15 s 95 oC and 30 s 60 oC. The relative expression ratio of targeted gene is calculated using the detected Ct value in comparison to a reference gene (housekeeping gene APRT (adenosine phosphoribosyl transferase)) by the standard method [70].

4. Some figures (Figs 2,3 and 5) do not have any Y axes legends.

We have added labels to the Y axes in all relevant figures with a description in the legends of Figs 2, 3 and 5 as “…were analyzed using relative ratio of gene expression by QPCR, and housekeeper gene APRT served as the reference gene.”

5. Ex vivo, in vitro used throughout the manuscript need to be italicized.

We have corrected this error throughout the manuscript.

Round 2

Reviewer 2 Report

I am satisfied with the revised version of the manuscript along with the responses provided by the authors.